# Radiation-Induced Heart Disease

**DOI:** 10.3390/jcm11010146

**Published:** 2021-12-28

**Authors:** Juan A. Quintero-Martinez, Sandra N. Cordova-Madera, Hector R. Villarraga

**Affiliations:** Department of Cardiovascular Medicine, Mayo Clinic College of Medicine and Science, Rochester, MN 55905, USA; quinteromartinez.juan@mayo.edu (J.A.Q.-M.); CordovaMadera.Sandra@mayo.edu (S.N.C.-M.)

**Keywords:** radiation therapy, echocardiography, cardiotoxicity, cardio-oncology

## Abstract

Cancer incidence and survivorship have had a rising tendency over the last two decades due to better treatment modalities. One of these is radiation therapy (RT), which is used in 20–55% of cancer patients, and its basic principle consists of inhibiting proliferation or inducing apoptosis of cancer cells. Classically, photon beam RT has been the mainstay therapy for these patients, but, in the last decade, proton beam has been introduced as a new option. This newer method focuses more on the tumor and affects less of the surrounding normal tissue, i.e., the heart. Radiation to the heart is a common complication of RT, especially in patients with lymphoma, breast, lung, and esophageal cancer. The pathophysiology is due to changes in the microvascular and macrovascular milieu that can promote accelerated atherosclerosis and/or induce fibrosis of the myocardium, pericardium, and valves. These complications occur days, weeks, or years after RT and the risk factors associated are high radiation doses (>30 Gy), concomitant chemotherapy (primarily anthracyclines), age, history of heart disease, and the presence of cardiovascular risk factors. The understanding of these mechanisms and risk factors by physicians can lead to a tailored assessment and monitorization of these patients with the objective of early detection or prevention of radiation-induced heart disease. Echocardiography is a noninvasive method which provides a comprehensive evaluation of the pericardium, valves, myocardium, and coronaries, making it the first imaging tool in most cases; however, other modalities, such as computed tomography, nuclear medicine, or cardiac magnetic resonance, can provide additional value.

## 1. Introduction

Cancer incidence and survivorship are rising; the global cancer statistics provided by the International Agency for Research on Cancer within the World Health Organization (WHO), which includes 185 countries around the world, estimate that, after 5 years of a cancer diagnosis (mainly breast, colorectal, and prostate), there is an estimated 51–71% survival rate. In the year 2020, cancer had an annual incidence of 19.3 million cases and was responsible for 10 million deaths [1,2]. Multiple treatment interventions have been developed in the urge to fight this deadly disease; RT is one of them, especially when combined with surgery and/or chemotherapy [3]. In the last decade of the 20th century, 20–55% of the patients with new cancer cases received RT as part of their treatment [4], and close to half of them were treated with a curative intent which contributed to an improved survival rate for this population [5,6]. 

The main goal of RT is to damage the genetic material of cancer cells, therefore inhibiting their growth and replication. This is accomplished by exposing the desired tissue to ionizing radiation which generates high-energy ions that deposit inside the cells, blocking their proliferation and/or inducing apoptosis [7]. One downside of this is the exposure of non-cancer “healthy” cells which suffer the same deteriorating effects. RT has improved over the years and now the number of healthy cells affected is significantly reduced; nevertheless, the risk of collateral damage to healthy tissues and organs is still an issue [8].

Radiation-induced heart disease (RIHD) is one of the major concerns when exposing patients to thoracic RT; this can occur years after treating diseases such as lymphoma, breast, lung, and esophageal cancer, to name the most frequent. Some of the increased risks of patients who receive RT when compared to the general population are the development of accelerated ischemic heart disease and valvular and pericardial disease [9,10]. For this reason, it is important to understand the pathophysiology of RIHD, the different techniques to improve radiation delivery while minimizing tissue damage, types of radiation, risk factors associated with RIHD, and how to diagnose early RIHD. 

## 2. History of Radiation Therapy

Great advances have occurred since 1898 when RT was first used. During its origins, physicians relied only on physical examination to determine the margins of the tumor. This had the downside of underestimating the tumor size and the progression of disease, and, for the subsequent follow-ups, making it difficult to define if there was tumor resolution in these patients. Another limitation in the early days of RT was the poor demarcation of 2D images to localize or differentiate tumors from soft tissue, deep tissue, or lymph nodes. Due to these factors, there was an inadequate estimation of the tumor burden. Additionally, the methods used to assess the direction and penetration of the radiation beams were inaccurate, making the tumor coverage and exposure to adjacent organs unpredictable [11]. Multileaf collimators were then introduced. This technology provided moving leaves to block some of the radiation beams that were directed to healthy organs; however, this approach was limited to geometrically shaped rectangular fields which were not accurate enough to cover the entire contour of the tumor and spare adjacent organs at the same time [12]. As efforts continued to improve, conventional 2D RT was replaced by 3D, conformal RT, which was first introduced in 1965 by Takahashi et al. [13]. Three-dimensional RT, as its name implies, uses three-dimensional axial-computed tomography to provide a precise delineation of the target tumor and adjacent organs. This new method was fundamental for the improvement of radiation dosage delivery. This led to novel intensity-modulated RT (IMRT); IMRT improves on 3D by modulating the dose delivered to each tissue to match the desired prescription goals [14]. Three-dimensional RT, IMRT, and dose-reducing radiation equipment were important for the standardization of protocols, and, nowadays, RT has reached a higher level of accuracy while minimizing collateral radiation to adjacent organs [15]. 

Another procedure introduced to improve RT delivery was the deep inspiration breath hold (DIBH); this technique is frequently used by radiation oncologists and has also been proven to significantly reduce radiation to the heart and lungs [16]. DIBH consists of using physiologic respiration to shift internal organs into a more convenient position which will ensure higher selective RT against the tumor cells and less to adjacent tissues. DIBH is performed by asking the patient to hold his breath up to the point when he achieves a determined abdominal elevation or height [17]. 

Several studies have proven DIBH to be very effective to reduce chest radiation. One study demonstrated that, when receiving breast/chest wall RT, all patients who used DIBH were able to meet their therapeutic goals of RT without surpassing the recommended daily radiation fraction doses for the heart. On the other hand, 56% of the patients without DIBH reached an unsafe level of radiation above the threshold [18]. Another study, including breast cancer patients who received left-sided IMRT, showed that the use of DIBH significantly reduced heart and lung radiation doses in six out of the nine patients. Additionally, the DIBH technique was able to completely avoid heart radiation exposure in two subjects [19]. 

Gray units (Gy) are used to quantify the ionizing radiation that deposits inside cells; it is defined as the absorption of 1 joule of radiation per kg [20]. High cumulative doses of radiation >30 Gy and high daily doses of radiation fractions >2 Gy are considered to increase the risk of RIHD [3,21].

One systematic review, which included 10 studies and a total of 268 patients, demonstrated that DIBH reduces heart radiation doses by 38–67% and, more importantly, left anterior descending coronary doses by 31–71% when compared to free breathing [22]. One of the biggest downsides of the DIBH is that it requires patient education and training for the procedure which can take up to 30 min per session. Additionally, it requires a greater number of staff members who need to be trained in this technique [23].

## 3. Pathophysiology of RIHD

Traditional RT (photon beam) affects the heart via micro and macrovascular mechanisms [3] which can lead to valvular disease [24], pericardial disease [25], conduction abnormalities [26], cardiomyopathy [27], and accelerated coronary artery disease [28]. One of the most understood pathophysiologic mechanisms is the macrovascular damage associated with the earlier development of age-related atherosclerosis [29]. This phenomenon is explained by endothelial damage that RT generates in the coronary arteries [30] which, consequently, causes an inflammatory response that releases a large number of cytokines responsible for macrophage activation and deposition of lipoproteins [31]. A similar mechanism to the formation of atherosclerotic plaques is seen in traditional coronary artery disease but in an accelerated fashion [32]. One retrospective study, which included 2168 women who underwent RT for breast cancer, found that their risk for major coronary events was increased by 7.4%; this increment begins 5 years after receiving RT and continues for 30 years. The risk was also higher for patients who received left- vs. right-sided RT. This study also proved that the risk increases with preexisting cardiac risk factors and higher radiation doses [28]. One systematic review, including six studies from 1996 to 2016, involved patients with low risk of CAD who received left breast/chest wall RT. Follow-up of the studies was limited to 6–12 months after RT. Four of these studies showed that cardiac exposure to radiation was associated with early myocardial perfusion defects [33]; these were mainly seen in the apical and anterolateral segments of the left ventricle (LV) and not associated with changes in the ejection fraction [34]. The same systematic review also proved that perfusion defects were strongly dose dependent and that patients who underwent cardiac radiation-sparing techniques, such as DIBH, had better outcomes [33]. In another study that included 7033 patients with Hodgkin disease who received chest RT, the risk for death from myocardial infarction was double when compared to the general population and it persisted for 25 years after treatment [35]. This risk was higher and independently associated in patients who were exposed to: (a) supradiaphragmatic total nodal RT (RR 9.0, CI 5.4–14.1), (b) mantle RT (RR 3.2, CI 2.3–4.2), (c) anthracyclines (RR 3.2, CI 1.9–5.2), and (d) vincristine (RR 2.0, CI 1.3–2.9) [35].

## 4. Types of Radiation Therapy

RT is utilized to eradicate targeted tumors and improve survival. There are two ways RT can be delivered: external beam radiation or internal radiation (also known as brachytherapy). This review will focus on external beam radiation; this form of RT is performed by targeting the tumor and delivering high-energy rays (photons, protons, electrons, neutrons) from outside of the body to the tissue. 

Traditional RT, also known as photon beam therapy, works by depositing energy inside the cells throughout the whole path of the beam; this can be inconvenient because the tissues surrounding the targeted tumor are usually affected, i.e., the heart. Newer therapies of energy rays involving particle beams (electrons, neutrons, and protons) have been studied in this field. Electron beams have a very low tissue penetration which is the reason why they have only proven to be useful for superficial skin diseases such as mycosis fungoides, cutaneous T-cell lymphoma, Sezary syndrome, Kaposi sarcoma, or inflammatory breast cancer, to name a few [36]. While this therapy has a low risk of collateral damage due to its poor penetration, the patients who will benefit from this intervention represent only a minority. Contrary to electrons, the neutron beams are very harmful by nature and are generally avoided in the clinical setting [37]. On the other hand, proton therapy has proven to provide the best dosage delivery to cancer cells with the lowest adverse effects to surrounding tissues. The energy being deposited by the protons can be steered so that it reaches a specific depth in the tissue and covers the entire length of the tumor (known as the Bragg peak effect) [38,39]. Proton therapy minimizes the radiation exposure of adjacent tissues to the tumor and allows to increase the doses of RT delivered to the cancer cells. This would otherwise be harmful when using photon therapy because of the higher risk of affecting adjacent tissues and organs [40,41]. However, the cost of proton beam therapy remains very high and is not affordable for every patient or institution, which is the reason why photon beam therapy remains a therapeutic option in many patients with cancer [42]. In some institutions, such as the Mayo Clinic, the additional costs of proton beam therapy are subsidized to allow the patient to receive the best treatment possible. 

Some studies in patients with breast cancer, Hodgkin lymphoma, and pediatric malignancies, such as medulloblastoma, have shown that there is a significant reduction in radiation doses to the heart with the use of protons over photons [43]. One systematic review that included 13 studies from 2002 to 2017 showed that proton therapy reduced the mean heart radiation dose by a factor of two- or three-fold in breast cancer patients when compared to traditional RT [40]. Another prospective study in breast patients, which evaluated the mechanical function of the LV by using 2D speckle tracking imaging, showed that the relaxation properties of the LV were compromised with photon but spared with proton therapy [44]. A systematic review including 14 studies in patients with Hodgkin lymphoma proved that proton therapy had a lower radiation weighted average when compared against traditional 3D RT (3.57 Gy less) and IMRT (2.24 Gy less) [45]. Finally, a study which compared proton vs. photon therapy in pediatric patients with medulloblastoma showed that the proton mean radiation heart dose was 0.2 Gy while photon was 10.4 Gy [46]. The adverse effects of proton beam therapy and its relationship with cardiotoxicity still require long-term follow-up in patients to be better defined, but, up to now, there is a definite decrease in the amount of radiation delivered to the heart [44].

## 5. Risk Factors for Reaching the Threshold of RIHD 

RIHD is associated with risk factors such as high radiation exposure, the use of other cardiotoxic medications (anthracyclines), prior history of heart disease, young age, and traditional cardiovascular risk factors. Patients who receive anterior left chest radiation were significantly associated with a higher risk of developing coronary heart disease (RR 1.29, CI 1.1–1.5) and cardiac death (RR 1.22, CI 1.08–1.37) when compared against those receiving right-sided RT [47]. High cumulative doses of radiation are associated with a higher risk of RIHD; one prospective study involving 2232 Hodgkin disease patients found that exposure to RT doses above 30 Gy increased their risk of cardiac mortality by 3.5-fold [48]. 

One study evaluated the risk of developing symptomatic congestive heart failure and myocardial infarction in 299 breast cancer patients who received chemotherapy +/− RT and then compared them against healthy controls from the Framingham epidemiologic study. The results showed that patients who received high doses (450 mg/m^2^) of anthracyclines concomitantly with left-sided RT had a tenfold increased risk of developing these cardiac events [49]. Young age was also proven to increase the risk of developing RIHD [50]; a study that evaluated patients younger than 25 years old who received RT showed a 7.5 times increased risk of developing coronary artery disease [51]. The presence of comorbidities, such as preexisting cardiac disease, diabetes, hypertension, obesity, and hypercholesterolemia, have also been associated with a higher risk of developing RIHD [52,53]. Genetic mutations in genes responsible for DNA repair pathways have been associated with increased radiation sensitivity and a higher risk of adverse effects [54]. 

## 6. Screening

Based on the above, it is essential to screen patients at risk for radiation-induced heart disease (RIHD). The best methods and frequency remain uncertain. Although guidelines/expert consensus for its evaluation have been published, the assessment and monitoring of heart function is similar to the standard procedures and tests cardiologists use in other patient settings [3,55].

Consequently, it is necessary for both radiation oncologists and cardiologists to recognize the risks and the underlying pathophysiology of RIHD [56]. Frequently, the evaluation required for these patients depends on different clinical scenarios, as detailed below:

Pericardial disease: electrocardiography (ECG), chest X-ray, and echocardiogram.

Myocardial dysfunction: echocardiogram, dobutamine stress echocardiography, 2D speckle tracking echocardiography (2D-STE), cardiovascular magnetic resonance (CMR), and computerized tomography (CT).

Valvular disease: echocardiogram, stress test (exercise, dobutamine stress echocardiography, Vo2 treadmill), and cardiac catheterization.

Coronary artery disease: ECG, echocardiogram, exercise stress test (non-imaging/imaging), and angiography (noninvasive, invasive).

Carotid artery disease: this topic is out of the scope of this review.

In this section, we will focus on the imaging tests available to evaluate these cardiovascular complications after RT.

### 6.1. Pericardial Disease

Pericardial disease (effusion and/or constriction) is a common complication of RT; its clinical presentation can be classified into acute or chronic disease [57]. The acute phase occurs days to months after RT, while the chronic phase may develop months to years later. Before the implementation of new methods to deliver RT, the incidence of pericarditis post RT used to be around 70% in patients with carcinomas (breast, lung, and esophagus), Hodgkin disease, and non-Hodgkin lymphoma (Hodking and All B-cell type NHL) [25,58]. In the last decades, the incidence has decreased to 6–30% but remains as the most common complication of RIHD [59,60]. A detailed history with physical examination is important for its diagnosis, but it is nonspecific and often requires an ECG and/or imaging studies to be able to differentiate constrictive from restrictive disease. Imaging can also offer information on mimics of constrictive pericarditis, such as pericardial tamponade, restrictive cardiomyopathy, right ventricular infarct/failure, pulmonary embolism, acute mitral regurgitation, or severe tricuspid regurgitation, providing a more accurate diagnosis [57].

#### Evaluation

Echocardiography is the first-line imaging method to evaluate patients with suspected or confirmed pericarditis [61]. This is due to its high sensitivity to detect anatomical and hemodynamic changes, especially in constrictive disease [62]. Contrast-enhanced CT and late gadolinium enhancement CMR can be used as complementary methods when needed; they can provide information on pericardial thickening, edema, or fibrosis (Figure 1) [63,64,65].

### 6.2. Myocardial Dysfunction

There are early complications of RT, such as inflammation, repolarization abnormalities, and mild myocardial dysfunction, but cardiotoxicity is more evident after 10 years, especially in patients who are exposed to doses above 30 Gy [21]. These chronic alterations include diffuse myocardial fibrosis with relevant systolic and diastolic dysfunction, conduction disturbances, and autonomic dysfunction [66]. All of them can contribute to LVEF abnormalities [55].

LVEF is an excellent predictor of myocardial systolic function; decreased values can be a surrogate of late cardiotoxicity. However, LVEF can be insensitive to detect subclinical or early myocardial dysfunction. These subclinical findings can be assessed with other methods such as 2D speckle tracking echocardiography (2D-STE) [67].

#### Evaluation

Echocardiography is essential to identify and monitor myocardial dysfunction. This method provides an accurate, fast, and noninvasive approach to measure LVEF and evaluate LV systolic and diastolic function [3,55,68].

LV systolic function: There are many techniques to evaluate the LVEF, but guidelines recommend using 2D volumes Simpson’s biplane methodology by echocardiography (a modality requiring area tracings of the LV cavity) as the first choice [69]. Two-dimensional echocardiography can evaluate global systolic function, and a drop below its normal range (<50%) can reflect RIHD. Nevertheless, subtle changes associated with early cardiac involvement are difficult to assess; some studies have demonstrated that 2D-STE is a good screening method to detect early changes in myocardial mechanical function [70,71]. Two-dimensional speckle tracking echocardiography evaluates myocardial function by measuring cardiac deformation throughout the cardiac cycle. Strain is analyzed in three different spatial domains of contractility (longitudinal, circumferential, and radial) and its main benefit in RT is its capacity for detecting myocardial dysfunction even when the LVEF is normal [68,72]. Global longitudinal strain is widely recognized and used for this matter, while circumferential and radial strain are reserved for research purposes. Strain rate is a more comprehensive 2D-STE technique that is mainly used for research and evaluates the same three spatial domains during systole (SRs) and early (SRe) and late diastole (SRa).

LV diastolic function: The evaluation of LV diastolic function is an integral component of the standard echocardiographic examination. The identification of these changes in the early phase of RT may be relevant, especially in breast cancer patients who have other risk factors for developing heart failure or are being treated with cardiotoxic medications [73]. LV diastolic function can be assessed using traditional echocardiographic parameters such as mitral inflow, tricuspid regurgitant velocity, or tissue Doppler of the mitral annulus and left atrial volume [74]. However, some studies have shown that these parameters are not able to detect early diastolic disfunction post RT [39,75].

Some studies have shown that 2D-STE may play an important role in detecting early diastolic dysfunction by measuring abnormalities in the early diastolic (SRe) and late diastolic strain rate (SRa) [39,44,75].

Sritharan et al. included 40 women with left-sided breast cancer undergoing photon RT with doses between 42.4 to 50 Gy. The investigators evaluated the diastolic function of these patients by measuring their diastolic strain rates at baseline, during RT, and 6 weeks post RT. Authors found that SRe and SRa were significantly reduced (longitudinal SRe (s−1)1.47+/−0.32 vs. 1.29+/−0.27; longitudinal SRa (s−1)1.19+/−0.31 vs. 1.03+/−0.24; *p* < 0.05) when comparing baseline vs. 6 weeks post RT [75]. Another study, which included 50 patients with breast or other thoracic cancers, evaluated conventional echocardiographic parameters and 2D-STE to predict myocardial dysfunction in subjects who received either photon or proton RT. While normal echocardiographic parameters were not able to detect differences between the two therapies, 2D-STE global circumferential, longitudinal, and radial SRe was abnormal in patients who received photon vs. proton beam therapy which was present even after 1 year’s follow-up [44]. These studies prove that early diastolic dysfunction is present after RT and that 2D-STE is a good tool to evaluate it.

### 6.3. Valvular Disease

Acute radiation effects in heart valves are commonly subtle, clinically less relevant, and challenging to assess. The incidence of radiation-induced valvular heart disease (RIVHD) is directly related to the dose of RT [76]. Additionally, RIVHD is time dependent; in a cohort of Hodgkin lymphoma patients, it was present in 1% at 10 years, 5% at 15 years, and 6% at 20 years [24]. Evidence suggest it is more common on the left side valves and changes consist of leaflet thickening, fibrosis, and calcification. RIVHD described 20 years post RT included mild mitral valve regurgitation in up to 48% of patients, mild aortic regurgitation in 45%, moderate to severe aortic regurgitation in 15%, mild to moderate aortic stenosis in 16%, and mild pulmonary regurgitation in 12% [3] (Figure 2).

#### Evaluation

Appropriate workup for patients with RIVHD includes a thorough clinical with physical examination and diagnostic testing.

Echocardiography is, again, the method of choice for the initial evaluation of these patients because it is highly sensitive in detecting any degree of valvular heart disease. It is also appropriate to evaluate for valvular thickening and calcification [77,78].

**Figure 2 jcm-11-00146-f002:**
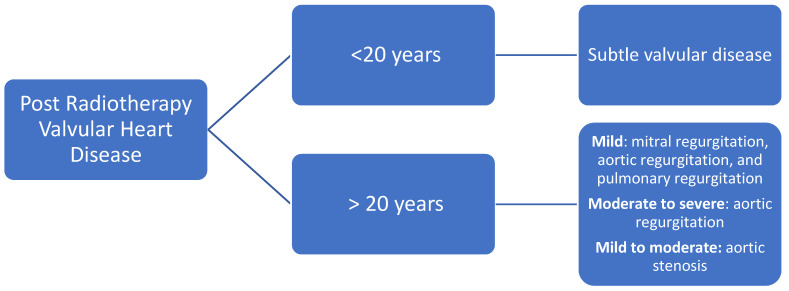
Most common types of valvular disease post radiotherapy [3,79].

### 6.4. Coronary Artery Disease

RT has been shown to be related with an increased risk of ischemic heart disease in patients with breast cancer and Hodgkin disease [24,51,80]. This risk is related to the accelerated atherosclerosis seen in RT which was described previously (Figure 3). One study performed in breast cancer patients suggested that CAD may develop as early as 5 years post RT [28]. Because of this, is recommended to evaluate and screen these patients before they develop significant disease. Screening will depend on the time they underwent RT and the patient’s risk factors.

#### Evaluation

The most useful methods to evaluate CAD include echocardiography, stress echocardiography, cardiac CT, CMR, and perfusion SPECT [81,82,83,84].

Echocardiography: Echocardiography is a valuable tool for assessing the cardiac structure and function in patients with coronary artery disease; it can determine the presence and extent of regional wall-motion abnormalities at rest which correlates well with CAD [82]. When this information is not sufficient to determine CAD, performing a test of inducible ischemia is recommended [85].

Stress echocardiography (exercise or dobutamine) has a low cost, minimal risk, and no exposure to ionizing radiation [86]. It is frequently used to evaluate patients for myocardial ischemia. It can recognize structural and functional alterations not evident at rest [87]; this method is highly sensitive and specific to detect abnormalities in the epicardial coronary arteries. Exercise testing is the recommended option because it can assess the physiologic functional performance. Heidenreich et al. included 294 asymptomatic patients with Hodgkin’s disease treated with mediastinal irradiation (≥35 Gy). Stress echocardiography (exercise and dobutamine) was used to evaluate CAD in these patients at baseline, 2–10 years, 11–20 years, and >20 years post RT. They found a prevalence of CAD of 7.5% 15 years after RT. Stress echocardiography had a positive predictive value of 80% to detect severe three-vessel disease and 87% for left main CAD after Hodgkin disease. The study also found that the presence of resting wall-motion abnormalities or ischemia on stress testing was associated with an increased risk of future cardiac events [81].

Coronary calcium score: calcium score is calculated based on the amount of calcification seen in the coronary arteries and matched with age and gender controls. This score has been extensively validated for detection of CAD in non-cancer patients [88]; nevertheless, its use post RT is not clear. One study evaluated 20 asymptomatic breast cancer patients who were less than 60 years old and were treated with RT at least 5 years prior to their enrolment. A CT calcium score was performed in all patients to assess its capacity to predict aortic and CAD. With a median interval between RT and CT calcium score of 8 years, this score was not able to detect significant atherosclerosis in either the aorta or coronaries [89]. Another study included nine Hodgkin lymphoma patients with an age between 35 and 60 years and performed a CT calcium score to evaluate their coronaries 12 years or more after their last RT. This study found six patients to have a calcium score above the 90th percentile for their age and sex group, suggesting that it might be a useful tool to evaluate RT-related CAD [83]. Larger prospective studies are needed to identify whether this score provides additional value against current evaluation methods in this population.

Coronary CT angiography has been used for follow-up in small groups of patients after RT for Hodgkin’s disease. These studies demonstrated advanced coronary calcification and advanced obstructive CAD in relatively young patients [83,90]. It is unclear whether CT can distinguish general atherosclerotic CAD from lesions caused by RT. In the absence of symptoms of CAD, there are currently insufficient data to recommend routine use of coronary CT angiography in patients who underwent high-dose RT.

Cardiac magnetic resonance (CMR) offers precise visualization of anatomical structures, assessment of cardiac function, and characterization of myocardial tissue, all within a noninvasive imaging modality [84]. T1 mapping can detect diffuse myocardial fibrosis and follow its changes over time. A study evaluated CMR images acquired during rest, adenosine stress, and late enhancement in 31 Hodgkin survivors 20 years after receiving RT. They found that CMR could detect reduced LVEF, hemodynamically relevant valvular dysfunction, late myocardial enhancement, and perfusion defects in approximately 70% of patients [91].

Radionuclide imaging (SPECT and PET) provides information on myocardial perfusion and wall-motion abnormalities that can be useful to follow-up patients who received RT and have a high risk of CAD [92]. Marks et al. [34] evaluated 114 patients treated with RT for left-sided breast cancer. A SPECT scan was used to assess myocardial perfusion, regional wall motion, and EF. It was performed at baseline, 6, 12, 18, and 24 months after RT. SPECT was able to identify an increased incidence of perfusion defects over time (27% at 6 months, 29% at 12 months, 38% at 18 months, and 42% at 24 months). It also detected wall-motion abnormalities in patients with perfusion defects vs. patients with no perfusion defects ((12 to 40%) vs. (0 to 9 %) *p* < 0.007).

### 6.5. Carotid Artery Disease

This entity is also described but is out of the scope of this review.

## 7. Surveillance Protocol after RT

### 7.1. Asymptomatic Patients

A baseline, comprehensive, echocardiographic evaluation is warranted in all patients before initiating RT to identify baseline cardiac abnormalities. During follow-up, a yearly history and physical examination with close attention to symptoms and signs of heart disease is essential.

In patients post RT who remain asymptomatic, screening echocardiography 10 years after treatment appears reasonable. In cases where there are no preexisting cardiac abnormalities, a surveillance transthoracic echocardiogram should be performed every 5 years after the initial 10-year echocardiographic screening examination [3] (Figure 4A).

In high-risk, asymptomatic patients (patients who underwent anterior or left-side chest irradiation with ≥1 risk factor for RIHD (the same as cardiovascular risk factors)), a screening echocardiogram may be advocated after 5 years of treatment. In these patients, the increased risk of coronary events 5–10 years after RT makes it reasonable to consider noninvasive stress imaging to screen for obstructive CAD [3].

### 7.2. Symptomatic Patients

The development of either new cardiopulmonary symptoms or new suggestive physical examination findings, such as a new murmur (mitral regurgitation, aortic stenosis, or regurgitation), should prompt transthoracic echocardiography and/or stress testing examination

Repeated stress testing can be planned every 5 years if the first exam does not show inducible ischemia (Figure 4B).

Patients can be evaluated with anatomical testing (CTA) or functional testing (exercise ECG, nuclear stress testing, or stress ECG) depending on the initial echocardiographic results and the clinical indication, as well as the local expertise and facilities [3].

## 8. Conclusions

Even though RIHD has decreased in the last decade, physicians must continue to evaluate for its presence in every patient exposed to heart radiation (especially in those receiving doses >30 Gy). It is advisable to screen these patients for traditional risk factors of coronary disease (obesity, hypertension, dyslipidemias, diabetes, smoking) and recommend appropriate lifestyle changes and/or treatment. The life expectancy obtained with anti-cancer therapy can be compromised by increased morbidity and mortality associated with cardiac complications such as pericarditis, myocardial dysfunction, coronary artery disease, and valvular disease.

In patients with cancer, echocardiography takes a crucial role in evaluating the morphology and function of the heart and represents the first imaging tool in most cases. Every patient that had RT must be screened with an echocardiogram 5–10 years post treatment depending on their risk factors (5 years post treatment for patients with high risk of RIHD and 10 years for patients with low risk). After this, patients must be screened every 5 years with a new echocardiogram/stress echocardiogram. Additional procedures that contribute to the assessment of cardiotoxicity after RT include 2D-STE (to diagnose early systolic and diastolic dysfunction), CT (to detect pericardial thickening, calcifications, epicardial coronary artery stenosis), CMR (to evaluate fibrosis), functional testing (exercise ECG, nuclear stress testing, or stress echocardiography (dobutamine or exercise)). Other methods, such as CT calcium score, may provide additional prognostic information.

## Figures and Tables

**Figure 1 jcm-11-00146-f001:**
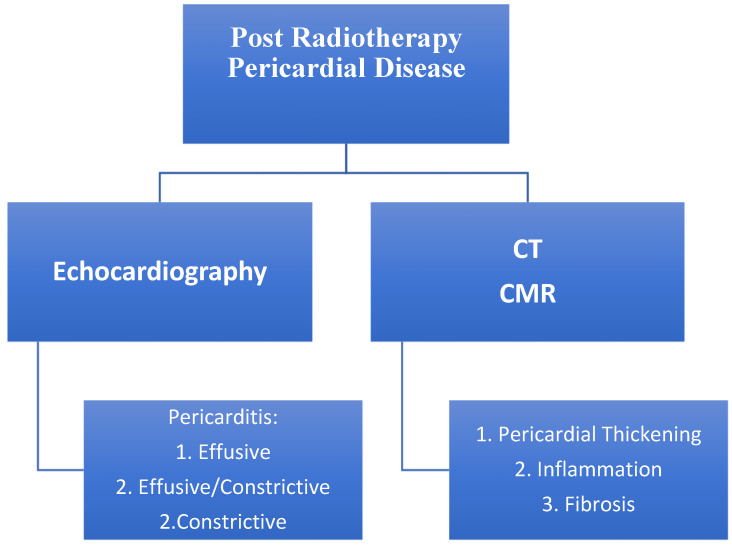
Evaluation of pericardial disease in patients who underwent radiotherapy. Abbreviations: CT: computed tomography; CMR: cardiac magnetic resonance [61,63,64].

**Figure 3 jcm-11-00146-f003:**
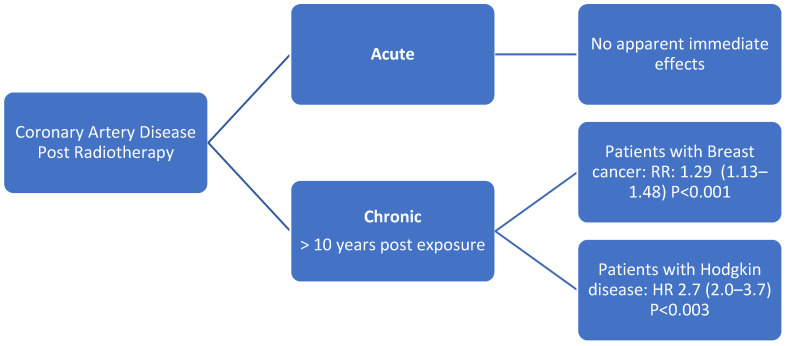
Radiation therapy and coronary artery disease [47,51,52].

**Figure 4 jcm-11-00146-f004:**
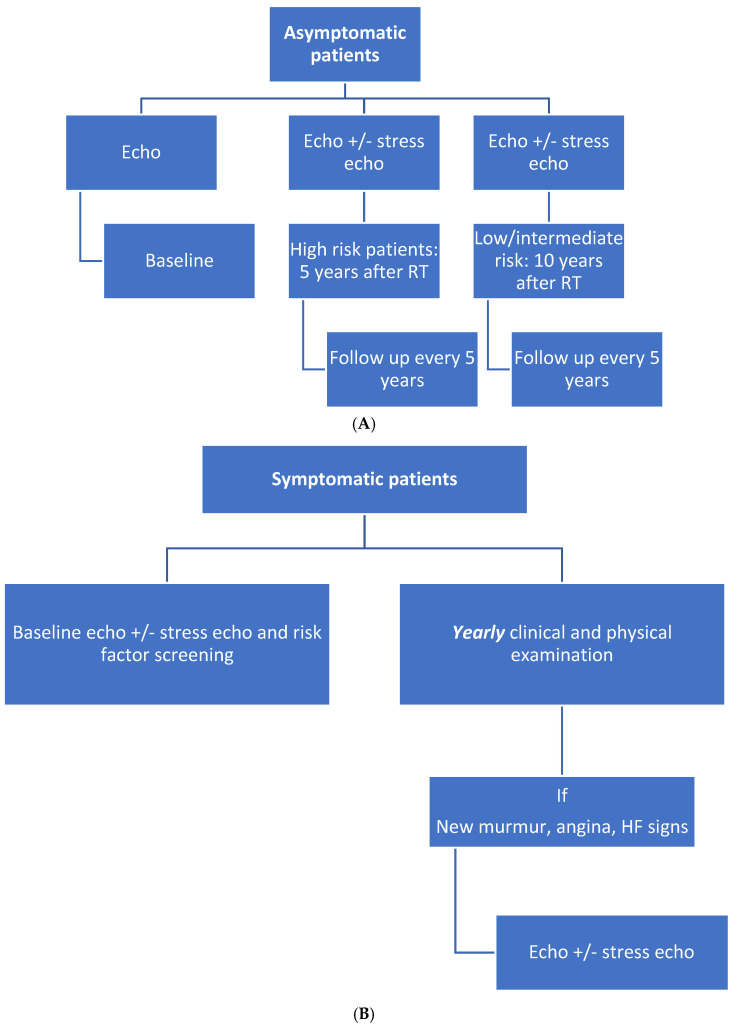
(**A**) Algorithm for management of asymptomatic patients after chest RT. (**B**) Algorithm for management of symptomatic patients after chest RT. [3] Abbreviations: Echo: echocardiography; HF: heart failure.

## Data Availability

Not applicable.

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
