# Peer review of "Radiation-Induced Heart Disease"

_jcm, 2021, doi:10.3390/jcm11010146_

Round 1

Reviewer 1 Report

Dear Authors,

I’ve read with interest your review Radiation Induced Heart Disease submitted for publication in Journal of Clinical Medicine.

The paper is well structured. The section 2 Radiation Induced Heart Disease and 4 Types of radiation therapy are particularly interesting

I’ve only some minor suggestions:

-There are some small misspelling in the figures.

-The sentence on the 3rd-4th line of page 4 probably should be put before (at the beginning of the paragraph)

-The description of strain analysis in the paragraph 6.2.1. (Evaluation: Echocardiography, both LV systolic and diastolic function) should be revised. The speckle tracking echocardiography and Doppler-based strain evaluation (strain, systolic strain rate (SRs), and early diastolic strain rate (SRe))” are different imaging modalities and should be distinguished (in the lines 12 of the paragraph 6.2.1.; in lines 6-7 and 18 of page 7 in PDF version).

Author Response

Dear Reviewers,

Thank you very much for your comments and feedback, this has significantly improved the quality of the manuscript. Please find the answer to your comments below in red.

Reviewer: 1

  1. There are some small misspellings in the figures.

Answer 1: Thank you for your comment, the misspelling in the figures has been corrected.

  1. The sentence on the 3rd-4th line of page 4 probably should be put before (at the beginning of the paragraph).

Answer 2: Thank you, we agree. This sentence was moved to the beginning of the paragraph and now it reads “RT is utilized to eradicate targeted tumors and improve survival. There are two ways RT can be delivered: external beam radiation or internal radiation (also known as brachytherapy) ……”

  1. The description of strain analysis in the paragraph 6.2.1. (Evaluation: Echocardiography, both LV systolic and diastolic function) should be revised. The speckle tracking echocardiography and Doppler-based strain evaluation “(strain, systolic strain rate (SRs), and early diastolic strain rate (SRe))” are different imaging modalities and should be distinguished (in the lines 12 of the paragraph 6.2.1.; in lines 6-7 and 18 of page 7 in PDF version).

Answer 3: We agree, this has been modified and reads “ Strain is analyzed in 3 different spatial domains of contractility (longitudinal, circumferential and radial) and its main benefit in RT is its capacity of detecting myocardial dysfunction even when the LVEF is normal.[68, 72] Global longitudinal strain is widely recognized and used for this matter while circumferential and radial strain are reserved for research purposes. Strain rate is a more comprehensive 2-D STE technique that is mainly used for research and evaluates the same 3 spatial domains during systole (SRs), early (SRe) and late diastole (SRa)”.

Reviewer 2 Report

The work is well written. It perfectly analyzes the toxic effects of radiotherapy also providing follow-up models.
I think that in assessing the presence of ischemic heart disease at 5-10 years after radiotherapy, the authors could further expand the role of provocative imaging tests. In particular, they could provide information on the role of echo-stress (see and cite for example the article by Novo et al published in JASE - Usefulness of Stress Echocardiography in the Management of Patients Treated with Anticancer Drugse) and stress cardiac magnetic resonance.

Author Response

Dear Reviewers,

Thank you very much for your comments and feedback, this has significantly improved the quality of the manuscript. Please find the answer to your comments below in red.

Reviewer 2

The work is well written. It perfectly analyzes the toxic effects of radiotherapy also providing follow-up models.
I think that in assessing the presence of ischemic heart disease at 5-10 years after radiotherapy, the authors could further expand the role of provocative imaging tests. In particular, they could provide information on the role of echo-stress (see and cite for example the article by Novo et al published in JASE - Usefulness of Stress Echocardiography in the Management of Patients Treated with Anticancer Drugse) and stress cardiac magnetic resonance.

Answer: Thank you very much for your comment. We have included additional information on the role of echo stress with references. It reads “When this information is not sufficient to determine CAD, preforming a test of inducible ischemia is recommended.[85] Stress echocardiography (exercise or dobutamine) has a low cost, minimal risk, and no exposure to ionizing radiation[86]; it is frequently used to evaluate patients for myocardial ischemia. It can recognize structural and functional alterations not evident at rest[87]”
